# The ΦPA3 phage nucleus is enclosed by a self-assembling 2D crystalline lattice

Eliza S. Nieweglowska [1], Axel F. Brilot [1,7], Melissa Méndez-Moran [1], Claire Kokontis[2], Minkyung Baek [3,4], Junrui Li[5], Yifan Cheng [1,5], David Baker [3,4,6], Joseph Bondy-Denomy [2] & David A. Agard [1]✉

To protect themselves from host attack, numerous jumbo bacteriophages establish a phage nucleus—a micron-scale, proteinaceous structure encompassing the replicating phage DNA. Bacteriophage and host proteins associated with replication and transcription are concentrated inside the phage nucleus while other phage and host proteins are excluded, including CRISPR-Cas and restriction endonuclease host defense systems. Here, we show that nucleus fragments isolated from ΦPA3 infected *Pseudomonas aeruginosa* form a 2-dimensional lattice, having p2 or p4 symmetry. We further demonstrate that recombinantly purified primary *Ph*age *N*uclear *En*closure (PhuN) protein spontaneously assembles into similar 2D sheets with p2 and p4 symmetry. We resolve the dominant p2 symmetric state to 3.9 Å by cryo-EM. Our structure reveals a two-domain core, organized into quasi-symmetric tetramers. Flexible loops and termini mediate adaptable inter-tetramer contacts that drive subunit assembly into a lattice and enable the adoption of different symmetric states. While the interfaces between subunits are mostly well packed, two are open, forming channels that likely have functional implications for the transport of proteins, mRNA, and small molecules.

There is a constant evolutionary pressure for bacteria to develop defense mechanisms against invading bacteriophages and for the phages to develop effective countermeasures[1]. To that end, restriction-modification and numerous CRISPR systems are widespread amongst bacterial hosts while phages have developed their own DNA modification and anti-CRISPR systems[2]. A subset of so-called jumbo bacteriophages, defined by having genomes exceeding 200 kb, have recently been shown to encode an elaborate system for sequestering the phage genome away from host nucleolytic attack, conveying broad resistance to DNA targeting by the host[3]. This is accomplished via the assembly of a selectively permeable protein shell that encompasses the phage genome[4,5]. The shell with its contents is referred to as the phage nucleus for its remarkable functional similarity to the eukaryotic nucleus: this structure forms a selectively permeable compartment that is centered within the host bacteria by a dynamic bipolar spindle formed from a phage-encoded, divergent tubulin called PhuZ[4,6–8]. The bacteriophage and host proteins involved in phage replication and transcription are concentrated within the phage nucleus shell while translation and nucleotide synthesis machinery, the aforementioned CRISPR-Cas and restriction endonucleases, as well as other host and exogenous proteins, are effectively excluded[3,4]. Unlike other proteinaceous prokaryotic compartments, such as carboxysomes or viral capsids, this shell can grow significantly throughout infection to reach nearly micron-scale, a process likely driven by genome replication[4,9–11]. In contrast to some other membraneless compartment-forming systems, such as protein condensates, subunits

[1]Department of Biochemistry, University of California San Francisco, San Francisco, CA 94143, USA. [2]Department of Microbiology, University of California San Francisco, San Francisco, CA 94143, USA. [3]Institute for Protein Design, University of Washington, Seattle, WA 98195, USA. [4]Department of Biochemistry, University of Washington, Seattle, WA 98195, USA. [5]Howard Hughes Medical Institute, University of California, San Francisco, CA 94143, USA. [6]Howard Hughes Medical Institute, University of Washington, Seattle, WA 98195, USA. [7]Present address: Sauer Structural Biology Laboratory, Center for Biomedical Research Support, University of Texas at Austin, Austin, TX, USA. ✉e-mail: agard@msg.ucsf.edu

do not diffuse throughout the shell of the phage nucleus, indicative of unique assembly properties[4,12].

While the total protein composition of the phage nucleus shell is unknown for any jumbo phage, Gp105 from the φKZ family phage 201φ2-1 was shown to be a marker for the shell[4]. It is the most highly expressed protein that is not part of the mature phage particle[4]. Here we formally introduce this protein and its related family members as *ph*age *n*ucleus *e*nclosure or PhuN proteins. The PhuN family currently includes Gp53 (φPA3), Gp54 (φKZ), Gp105 (201φ2-1), Gp202 (PCH45), and numerous putative homologous proteins from newly sequenced jumbo bacteriophages[4,5,13,14]. Beyond these other phage proteins, PhuN has no clear previously characterized homologous relatives. This unique assortment of biophysical and biological properties, combined with the mystery of the tertiary structure of PhuN, established it as an exciting target for structural and biochemical analysis.

In this work, we demonstrate that phage nucleus fragments isolated from φPA3 infected *P. aeruginosa* cells form a quasi-square lattice. We further show that Gp53, the PhuN family member from bacteriophage φPA3, readily assembles into large 2D lattices in vitro. We utilized a limited tilt, cryo-electron microscopy data collection scheme paired with single particle processing to determine a 3.9 Å map of PhuN assemblies. Assisted by RoseTTAFold[15] and AlphaFold[16], we present an atomic model and analysis of the assembly enclosing the phage nucleus. The structure reveals a two-domain core that assembles into tetramers. Flexible termini and large loops mediate adaptable inter-tetramer contacts that drive shell assembly into a largely p2 symmetric lattice. While the interfaces between subunits are mostly well-packed, two of the interfaces are open, forming clear channels that likely have important functional implications. Despite the lack of detectable sequence homology, analysis of the atomic structure reveals that the basic architecture is derived from a fusion of acetyltransferase and nuclease domains.

## Results

### Isolation of phage nucleus fragments

*P. aeruginosa* were infected with φPA3 and lysed after 60 min of infection. The lysates were separated using differential centrifugation to isolate large fragments of the phage nucleus. The resulting isolates resemble full or fragmented shells and have a clearly defined lattice structure (Fig. 1a). Two-dimensional averaging of the isolated fragments further reveals a lattice with subunits of both p2 and p4 symmetry with unit cell lengths and angles of 120 Å, 120° and 110 Å, 90°, respectively (Fig. 1a). The subunits forming the lattice appear to be a single, ~60 by 40 Å protein species. Recent low-resolution in situ tomography of phage nuclear shells from phages 201φ2-1 and Goslar reported p4 symmetry but is otherwise consistent with this observation, demonstrating a lattice with similarly sized subunits[17]. The isolated shell fragments often appear folded over and the lattice can vary in directionality, suggestive of both structural plasticity and imperfections as well as locally altered symmetry within the lattice of endogenous mature phage nuclear shells (Fig. 1a).

### PhuN self-assembles during purification

φPA3 PhuN is a 66.6 kDa protein with marginal solubility upon expression in *E. coli*, a problem surmounted by the addition of an N-terminal maltose-binding protein (MBP) solubility tag. Importantly, the addition of MBP does not prevent PhuN from incorporating into the phage nuclear shell during φPA3 infection and fragments isolated from MBP-PhuN expressing *P. aeruginosa* after φPA3 infection exhibit a lattice similar to that composed of only wild-type subunits (Supplementary Fig. 1a–c). All in vitro data presented here utilize the MBP-PhuN fusion and, for simplicity, will generally be referred to as PhuN.

During purification, exposure to an anion exchange column induced the formation of a broad spectrum of PhuN assemblies

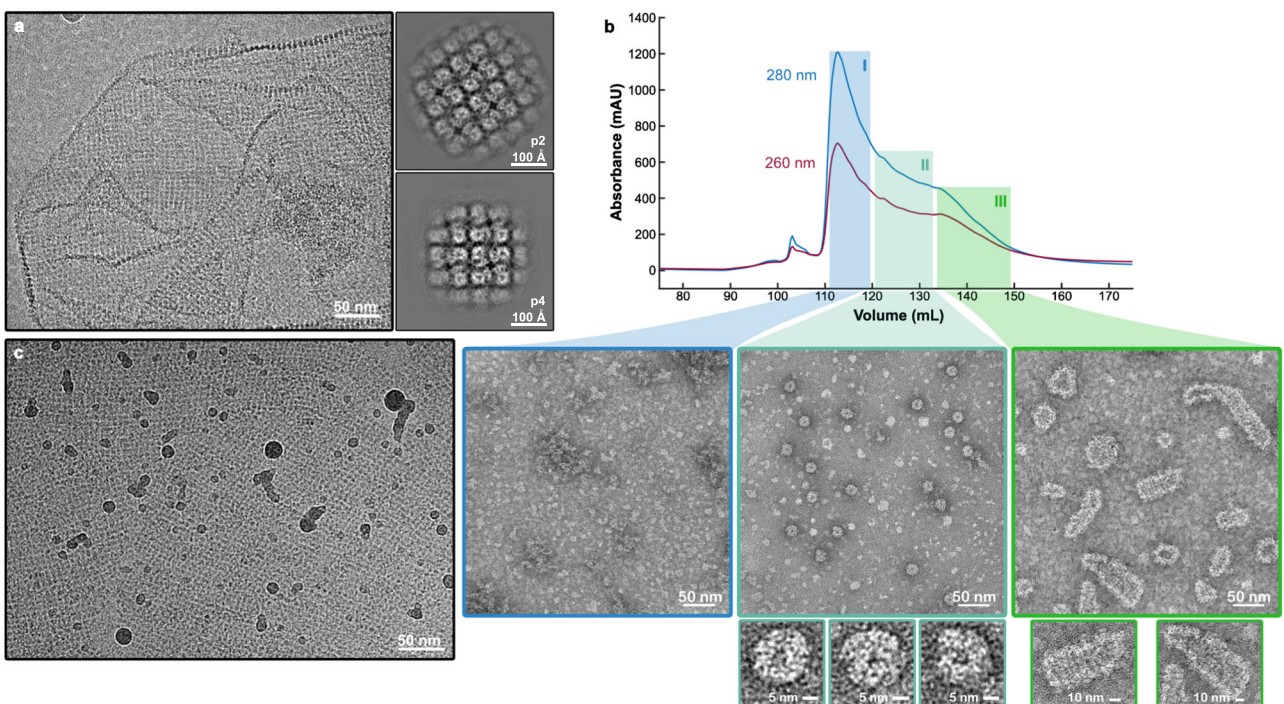

**Fig. 1 | PhuN forms a broad range of assemblies. a** Micrograph showing an unlabeled phage nuclear shell fragment isolated from ΦPA3 infected wild-type *Pseudomonas aeruginosa* alongside the subsequent p2 (120 Å, 120° unit cell length, angle) and p4 (110 Å, 90° unit cell length, angle) symmetries observed upon 2D classification. Folds and differences in lattice orientation are visible. (*n* = 1 unlabeled shell isolation experiment). **b** Anion exchange trace showing extended elution profile and corresponding negative stain EM micrographs. Class I (blue) shows concentrated monomeric species. Class II (turquoise) shows medium-sized assemblies in the 25 nm range. Class III (green) reveals assemblies with elongated shapes and striated textures. **c** In vitro assembled 2D crystalline arrays at 0° tilt showing three main patches with distinct lattice orientations and gentle curvature. Borders between different crystal patch orientations appear smooth. (*n* = 6 grids imaged for this publication).

observed upon elution (Fig. 1b). While the exact reason for this preferential on-column assembly is not known, it likely occurs as a result of increased local PhuN concentration during sample application, binding, and elution. Exposure to the high surface charge density of the column resin may also play a role.

These assemblies were grouped into three broad classes based on their appearance in negative stain electron microscopy. Class I primarily contains monomeric PhuN (Fig. 1b). Class II includes rounder species ~25 nm in diameter with distinct edges. Despite efforts to optimize the production of these for structure analysis, they proved to be an elusive species, often outnumbered by more irregular assemblies like those found in Class III (Fig. 1b). Class III consists of highly variable 50–130 nm long, elongated assemblies (Fig. 1b). The Class III species revealed a striated organization highlighting the striking diversity in both the shape and size of PhuN assemblies. In cryo-EM, although heterogeneously shaped, Class III assemblies appear to have a more regular, lattice-like organization and showed a density around the exterior consistent with the presence of the N-terminally fused MBP (Supplementary Fig. 1d, e).

### Growing and imaging PhuN 2D lattices

We initially set out to determine conditions that would allow us to control the assembly rate and state of PhuN. Using the anion exchange column as an assay to monitor shifts in assembly states, we instead found that at pH 6.5 monomeric PhuN forms large, 2D polycrystalline assemblies when applied to negatively charged grids for negative stain EM (Supplementary Fig. 2b). The low pH was also necessary to maintain the integrity of shell fragments isolated from infected cells, suggesting that pH may be an important factor to consider in shell assembly in vivo. Stress induced by antibiotic exposure has been observed to decrease the cytosolic pH to 6.7 in *P. aeruginosa* and other bacteria[18]. Perhaps phage infection may act as a similar metabolic stressor and have an analogous impact on pH.

These 2-dimensional crystals were further optimized using a 2D surface crystallization method as often used in 2D electron crystallography[19]. Purified His-tagged PhuN was diluted into a buffer droplet in Teflon wells that had a pre-formed lipid monolayer containing 21% Ni-NTA modified lipids (Supplementary Fig. 3). This approach yielded larger, better-ordered crystalline assembly monolayers than both those observed in negative stain and those isolated from infected cells. These arrays were then prepared for and imaged by cryo-EM. On conventional grids, few lattice patches could be found and those that did adsorb were only visible in very thick ice. We overcame this challenge by utilizing amino-functionalized graphene oxide grid supports[20] for the bulk of our data acquisition as they stabilized the lattices in somewhat thinner ice and greatly increased the number of lattice patches adsorbed. Despite optimizing lattice formation, the 2D sheets generally included many local regions of distinct lattice orientation as well as numerous bends and waves (Fig. 1c and Supplementary Fig. 4). Consequently, we chose a predominantly real-space approach, analogous to single particle analysis[21], instead of the more traditional 2D electron crystallography strategy (Supplementary Fig. 5). To obtain the necessary views, data were collected at multiple tilt angles: 0°, 15°, 30°, 35°, 40°, 45°, 50°, 60° (Supplementary Figs. 4, 5). In order to compensate for the high degree of beam-induced motion and lower image quality resulting from increased sample thickness in our tilted samples, data were collected at a high frame rate and processed using a tilt-optimized version of MotionCor[22]. Using this approach, we were able to resolve PhuN assemblies in 2D arrays to ~3.9 Å and build an atomic model starting with predictions from RoseTTAFold[15] and AlphaFold[16] (Fig. 2b–d, Supplementary Figs. 5, 6, and Supplementary Tables 1, 2).

### PhuN lattices: four unique interfaces

The φPA3 PhuN monomer is comprised of two connected domains. Despite the lack of convincing sequence homology to other proteins, structural comparison of our model with coordinates in the PDB using the Dali Structure Comparison Server[23] revealed that the large domain is structurally homologous to acetyltransferases while the smaller domain has a less clear fold homology, sharing similarities with proteins including GTPases, transport proteins, as well as several restriction endonucleases (Supplementary Table 3). As other proteins forming nanocompartments or encapsulins often share structural similarities with capsids[24], the PhuN homologies suggest a unique evolutionary origin. Since no related binding sites or catalytic residues could be identified in PhuN, it is unlikely that either domain has retained any ancestral catalytic activity. It remains to be seen whether these homologies have functional implications beyond a purely structural role.

PhuN assembles into a tetrameric lattice with a dimeric asymmetric unit, giving rise to p2 symmetry with a unit cell length and angle of 120 Å and 100°, respectively (Fig. 2a). At the 2D classification stage, we observed a minor subset of p4 symmetric particles (unit cell length and angle: 120 Å, 90°), as in the isolated phage nuclear fragments (Supplementary Fig. 7a). Alignment of the p2 asymmetric subunit models using the acetyltransferase-like domain results in an RMSD of ~2 Å for all Cα and 0.77 Å for the core Cα. The bulk of the differences between the structured domains of asymmetric subunits appears to arise from small shifts in helices as well as interconnecting loops. Each core tetramer is held together by the tight packing of four monomers with both domains contributing, while interactions between the tetramers are looser and predominantly coupled via highly flexible elements (loops, termini, etc.). In keeping with the p2 symmetry, each PhuN tetramer forms four unique interfaces, two of which form channels: a Diamond channel at the center of the core tetramer, an Open Hairpin channel and a Closed Hairpin interface forming the lateral contacts between core tetramers, and a Loop interface where the outer corners of four tetramers meet (Fig. 2a, e).

The central Diamond channel is lined by the four smaller PhuN domains of the core tetramer. The resulting diamond-shaped opening has a highly negative charge (Fig. 3b) and measures 32 Å by 33 Å across the corners. In an elegant domain swap that holds the subunits tightly packed around the channel, a small positively charged helix within the N-terminal 37 residues of the nuclease domain threads into the neighboring subunit, nestling at the top within a negatively charged pocket (Fig. 3a).

The Open Hairpin channel and Closed Hairpin interface are located at the lateral interface of two core tetramers. These lateral interactions are established by the same β-hairpins making contact with different residues of the neighboring tetramers, approximately 20 Å apart (Fig. 3c and Supplementary Fig. 8a–d). At the Open interface, the tips of opposing β-hairpins (residues 111–126) interact with residues near where the C-terminus leaves the structured acetyltransferase-like domain of the neighbors (Supplementary Fig. 8a, b, d). Keeping these hairpins apart creates a somewhat narrow channel, measuring 31 Å by 41 Å across the corners of the channel. At the Closed interface, these opposing β-hairpins form direct lateral interactions that close the channel and move the tips of the β-hairpins closer to a structured loop (residues 245–258) (Fig. 3c and Supplementary Fig. 8c–d). As we do not observe any large conformational changes between the asymmetric subunits, this suggests that the 20 Å shift between the β-hairpin positioning at the Open versus Closed Hairpin interfaces arises from a largely rigid motion of the entire core tetramer and thus, as a result, introduces the 10° skew in the lattice network (Fig. 2a and Supplementary Fig. 8e, f). One conjecture is that perhaps these are not fixed interfaces, but rather represent Open and Closed states that the lattice can adopt for a functional purpose such as relieving tension in the nearly micron-scale assemblies or adjusting the channel created at the

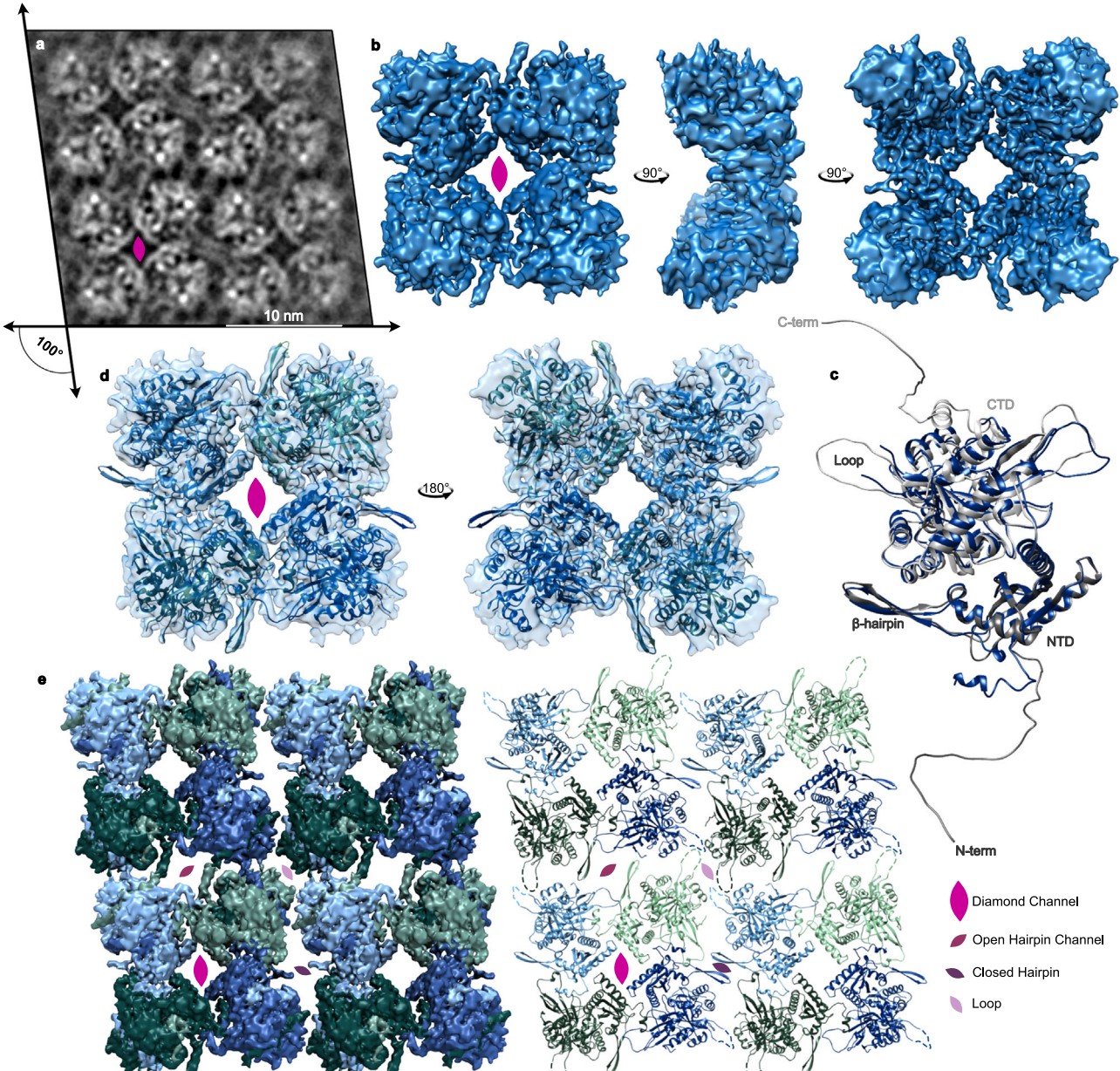

**Fig. 2 | PhuN assembles into a p2 symmetric lattice. a** Example 0° tilt 2D class average highlighting the p2 symmetry and resulting in four unique interfaces. The lattice is skewed to an -100° angle. The positioning of the flexible loops that are difficult to resolve in 3D is visible. **b** Highest resolution map of the core tetramer centered on the Diamond channel as viewed from different angles. **c** The final PhuN-C subunit model (blue) overlaid with the AlphaFold prediction used as our starting model (gray). Residues 1–18, 276–287, and 556–602 had no corresponding density in our map and were deleted from the model. **d** The tetrameric model of PhuN-C and PhuN-O fit into the map in **b**. **e** The map from **b** is segmented and positioned to recreate the channels observed in 2D classes (left) paired alongside the corresponding 16-mer-model (right). In both the segmented maps and models, PhuN-O is represented in green while PhuN-C subunits are represented in blue. Unresolved and unmodeled loops are represented with dashed lines to reflect the 2D class while the C-terminal tails are excluded.

Open interface (Supplementary Fig. 8e, f and Supplementary Movie 1). Going forward, we will refer to the asymmetric subunits by the positioning of their β-hairpins such that the β-hairpin from PhuN-O forms the Open channel while the PhuN-C β-hairpin forms the Closed interface.

The Loop interface appears to be, in part, held together by interactions between the large flexible loops (residues 272–291) from at least two different subunits. While the full densities of these loops remain unresolved in our 3D map, they are clearly visible in 2D classes (Fig. 2a). The loops from two PhuN-O subunits from opposing core tetramers interact laterally (Fig. 2a). The loops belonging to the PhuN-C subunits at those interfaces are not visible suggesting they are

unstructured or in an alternative, less visible conformation in the 2D class. These loops account for the largest conformational differences we observed between asymmetric partners.

The final 12 residues of the C-terminal tail reside in a groove between the two domains of PhuN, right over the β-hairpin (Fig. 2e). We also see strong densities in our high-resolution map for what are likely the C-terminal strands at the back of the acetyltransferase-like domain (Fig. 2e). These densities differ in the two asymmetric subunits. Unfortunately, there was insufficient local resolution to trace how the C-terminus crosses between tetramers. This leaves some ambiguity as to which exact paths the long termini take to cross the Open Hairpin channel and Closed Hairpin interface before resting their

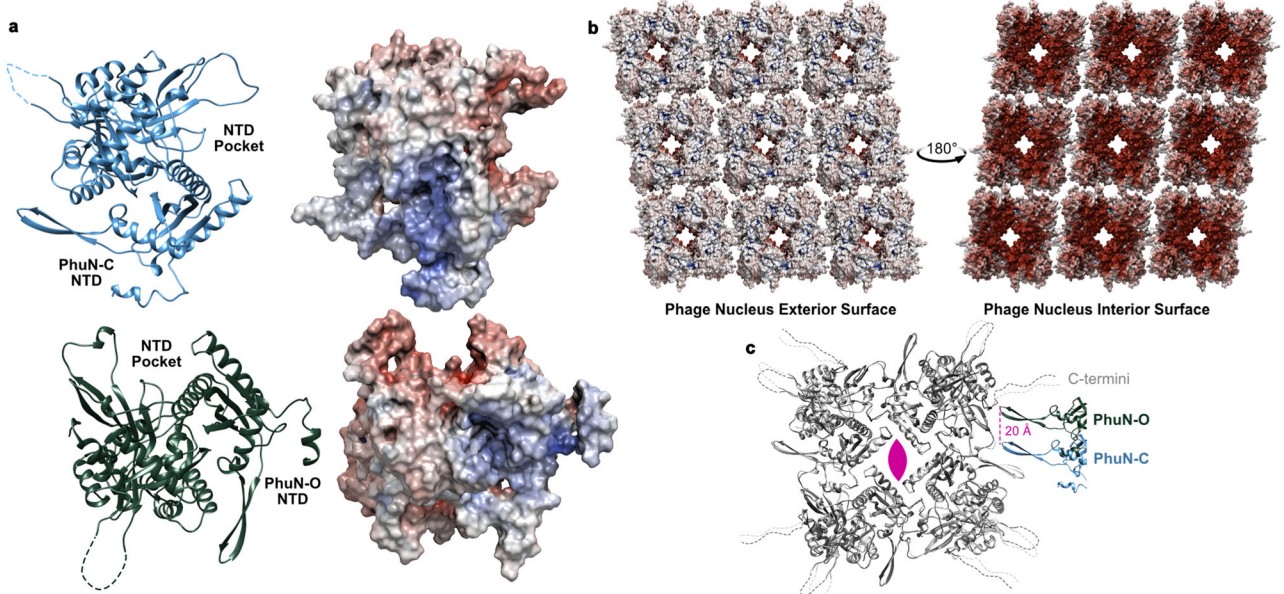

**Fig. 3 | A closer look at PhuN interactions and electrostatic potential. a** Surfaces of PhuN-C (top) and PhuN-O (bottom) colored by electrostatic potential with their corresponding models on the left. **b** Electrostatic potential coloration of the interior and exterior phage nucleus surfaces created by PhuN assemblies. Surface charge calculation was done at pH 6.5 and includes the C-terminal tail tips. Red corresponds to negative and blue corresponds to positive charge. **c** Direct comparison of PhuN-C and PhuN-O. The asymmetric subunits in the tetramer are overlaid showing minor differences (light and dark gray) while the interaction with the neighboring β-hairpins differs dramatically, shifting by ~20 Å from near the C-terminus (PhuN-O) towards a flexible loop (PhuN-C).

final 12 residues in the positively charged groove (Fig. 2e). One possibility is that the unresolved C-terminal tail regions may be free and unstructured to allow flexibility for the tetramers to accommodate alternating between Open and Closed Hairpin states, shell curvature, or different lattice orientations and symmetry states.

## PhuN symmetry and assembly

Since we observe the Loop, β-hairpin, as well as the N- and C-terminal tails at the assembly interfaces, we tested deletions of these regions for their ability to self-assemble in vitro and their ability to integrate into phage nuclear shells in vivo during φPA3 infection, as compared to full-length PhuN (Fig. 4a). The in vitro samples were fused to MBP as before and the in vivo deletions were fused to mNeonGreen for live fluorescence microscopy. Deleting the 37 N-terminal residues abrogated PhuN incorporation into the phage nuclear shell in vivo and resulted in no clear assembly in vitro (Fig. 4b and Supplementary Fig. 9b). This suggests that the N-terminal residues are necessary for the formation of the core tetramer and further assembly. In vitro, deleting the C-terminal tail severely compromised both assemblies as seen by anion exchange (Fig. 4f) and lattice formation as seen by negative stain EM (Fig. 4d and Supplementary Fig. 9d). In contrast, incorporation into the phage nuclear shell in vivo was minimally affected (Fig. 4d and Supplementary Fig. 9d), perhaps due to the presence of wild-type protein and multivalent C-terminal interactions of PhuN in the lattices. In a rather surprising result, deletion of the β-hairpin removed the ability of PhuN to integrate into the phage nuclear shell during infection, yet the purified deletion still formed 2D lattices as observed in negative stain EM (Fig. 4e and Supplementary Fig. 9e). This suggests that, while PhuN does not necessarily require the β-hairpin to form lattices, assemblies lacking the β-hairpin are biophysically, biologically, or otherwise unfavorable in the cellular context in the presence of wild-type protein. The deletion of the large Loop (Fig. 4c and Supplementary Fig. 9c) significantly decreased but did not abolish PhuN integration into the phage nuclear shell during infection. The Loop deletion similarly showed a mild propensity for both lattice-like assembly and enrichment of the small 25 nm Class II species we initially observed with the

full-length protein but no long-range order in negative stain (Fig. 4c and Supplementary Fig. 9c). These deletions together show that the Loop, β-hairpin, as well as the N- and C-terminal tails we observe at the interfaces of intra- and inter-tetramer interactions do, in fact, play critical roles in the ability of PhuN to assemble into lattices and form the phage nuclear shell in vivo.

During the review of this manuscript, a publication reported in vitro assembled four-fold symmetric cuboids of the homologous PhuN protein from bacteriophage 201φ2-1[17]. Despite the clear difference in observed symmetries, the core domains of the 201φ2-1 monomers closely match the φPA3 monomers we describe in this work (Fig. 5a). By contrast the quaternary organizations of monomers within the tetramer differ by an out of plane rotation of ~64°, corresponding to the large difference between the flat sheet subunit arrangement in our arrays and the cuboid organization described for 201φ2-1 (Fig. 5c–e). Accommodating this different organization requires alterations in a loop (residues 242-259 φPA3 and 271-289 201φ2-1), near the β-hairpin (residues 106–129 φPA3 and 134–157 201φ2-1), and in the N-terminal tail (residues 19–36 φPA3 and 48–64 201φ2-1). While the N-terminal helix and binding pocket are analogous to that of our φPA3 PhuN, the 201φ2-1 N-terminal tail is extended further away from the core domains of the protein to form the faces of the in vitro cuboids (Fig. 5a, b). Moreover, the tetrameric φPA3 model from our in vitro assembled lattices has an excellent fit into the 24 Å cryo-electron tomography density[17] obtained from 201φ2-1 infected cells (Fig. 5c–e) (masked correlation coefficients: 0.69 φPA3 tetramer; 0.49 calculated and 0.56 reported[17] for the 201φ2-1 cuboid tetramer). This observation, along with the agreement between our isolated and in vitro assembled lattice 2D classes, indicates that the subunit arrangement in our tetrameric model is an accurate representation of the lattices formed in vivo by PhuN family proteins. These two distinctive assemblies—highly constrained p4 cuboids[17] and p2/p4-containing flat sheets—formed by proteins with nearly identical tertiary structures highlight the remarkable adaptability of the PhuN protein family and invite future investigation of the p2 and p4 symmetries.

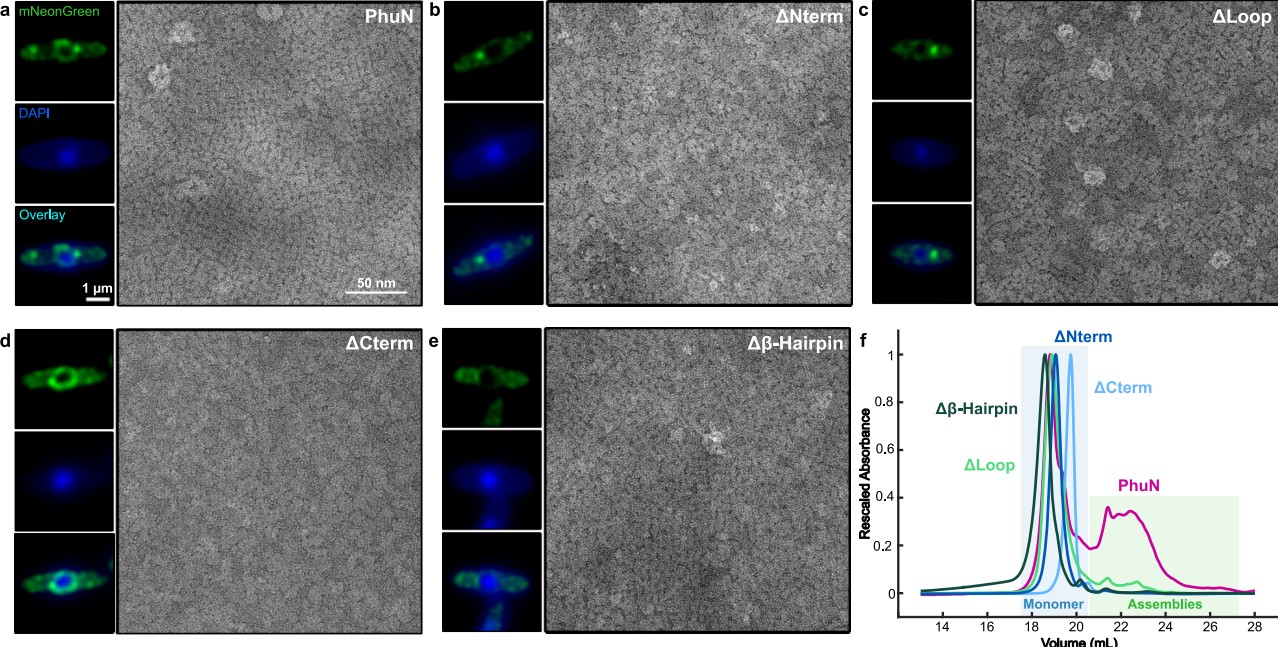

**Fig. 4 | φPA3 PhuN deletions show defects in shell integration in vivo and self-assembly in vitro.** Live fluorescence microscopy and negative stain EM micrographs of **a** full-length PhuN as well as the following PhuN deletions: **b** N-terminal tail (residues 1–37), **c** Loop (residues 272–291), **d** C-terminal tail (residues 556–602), and **e** β-Hairpin (residues 111–126). Fluorescence microscopy images of PhuN or PhuN deletions were collected at 40 min post-infection, deconvolved, and are displayed in green while the DAPI-stained DNA is shown in blue. The 1 μm and 50 nm scale bars apply to all fluorescence and EM panels, respectively. (Independent sample preparation and imaging: fluorescence microscopy n = 3, negative stain EM n = 1). **f** Rescaled anion exchange traces for the deletions shown in panels **a**–**e**. The decreased propensity for PhuN assembly on the column is evidenced by the sharp increase in the primary peak corresponding to the monomeric species (blue box) relative to the assembly peaks (green box). Samples from monomeric peaks were used for negative stain in panels **a**–**e**.

To better understand where the different symmetries originate in our samples, we mapped our p2 and p4 particles back to their micrographs and found they often reside in distinct clusters on the lattices (Fig. 6a). These clusters suggest that the different symmetries may occur in regions distributed around the phage nucleus in infected cells, perhaps serving functional roles such as enabling shell curvature, flexibility, and assembly or enabling the passage of small molecules. Given the similarity of the p2 and p4 core domains, honing in on the unresolved Loop, C-terminal tail, and interfaces between tetramers will likely be key to understanding the functional role of the different symmetries and ultimately delineating the assembly mechanism of the phage nuclear shell.

While we do not observe the N-terminal MBP tag in our maps, the MBP appears on the exterior of our small in vitro assemblies (Supplementary Fig. 1d). From this, we speculated that the lattice surface with the exposed N-terminus faces the cytosol. This is the same orientation suggested by the excellent fit of our tetrameric model into the 201φ2-1 nuclear shell map[17] determined by in situ tomography. Given this orientation, the exterior of the phage nuclear shell is sporadically positively charged while the interior surface is quite negatively charged (Fig. 3b). This could be an elegant strategy for keeping the DNA away from the surface, preventing it from inadvertently leaving the protection of the lattice network.

## Discussion

Once thought to be a defining feature of eukaryotic organisms, the phage nucleus proves to be a protein-based, membrane-free structure accomplishing a similar task. In this work we describe the self-assembly of the 66.6 kDa PhuN protein from bacteriophage φPA3 into a predominantly p2 symmetric lattice that closely matches fragments of the endogenous phage nucleus isolated from φPA3 infected *P. aeruginosa* as well as in situ reconstructions from the closely related bacteriophage 201φ2-1[17]. This tetrameric lattice features four distinct interfaces,

two of which form clear channels (Diamond, Open Hairpin) that could allow the passage of small molecules, unfolded proteins, or nucleotides. We further show that the lattice assembles through interactions between flexible loops and establishes a negatively charged interior surface for the phage nucleus.

These lattices and channels provide a framework with which to approach further mechanistic questions in this system ranging from growth and adaptability of the lattice, to DNA protection and packaging, to selective passage in and out of the phage nucleus. The channels we identify may allow for the diffusion of small molecules as is or perhaps can accommodate larger molecules following conformational changes induced at specific sites or in response to the binding of other phage factors. The different interactions of the large flexible loops and the possibility for varied C-terminal tail binding observed here could help accommodate such changes. This idea is supported by both our observation of a minor population of p4 symmetric particles in our lattices (Supplementary Fig. 7a) and the recent report of p4 symmetric cuboids assembled in vitro using 201φ2-1 PhuN[17]. The fact that the same monomeric subunit can readily form distinct PhuN symmetry states and assemble into structures ranging from flat sheets to highly constrained cuboids, as seen in 201φ2-1, implies an unexpectedly flexible, dynamic assembly far different from the highly stereotyped, rigid lattices found in viral capsids or compartments like the carboxysome[10,11,25,26]. In vivo, scattered throughout this remarkable lattice may be other protein-based channels to allow for capsid docking, efficient DNA extrusion, and packaging into empty capsids, as well as the selective passage of other large molecules.

In many other protein-based assemblies, such as eukaryotic and prokaryotic viral capsids and bacterial microcompartments, space is enclosed through combinations of triangular or hexameric units forming flat faces and pentameric units providing curvature[10,11,26]. At the molecular level, the ability of the same proteins to pack in these different modes arises from quasi-equivalent interactions amongst

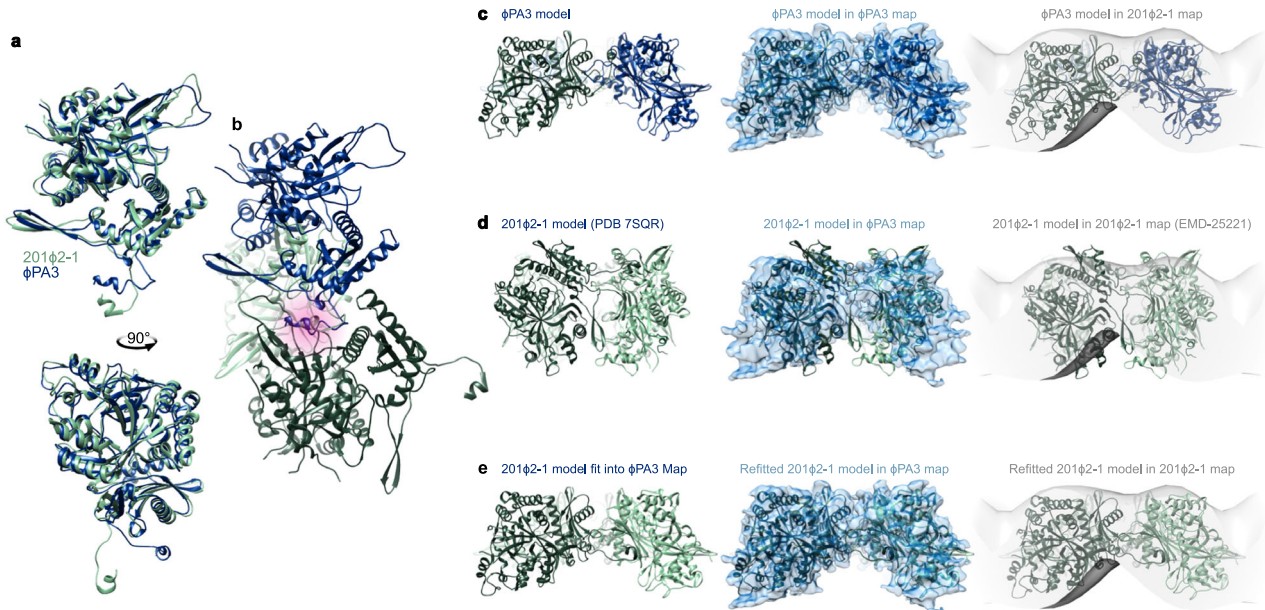

**Fig. 5 | PhuN proteins from φPA3 and 201φ2-1 share similar structures. a** Top and side views of an overlay of the φPA3 (blue) and 201φ2-1[17] (green) monomeric subunits. The overlay was determined by fitting the monomers into the blue φPA3 EM density shown in panels **c–e**. The greatest backbone differences between the models reside in the extended positioning of the N-terminal residues, β-Hairpin, and a resolved flexible loop. **b** The binding pockets utilized by both φPA3 and 201φ2-1 proteins are the same, as highlighted in pink. This was determined by fitting the dark green 201φ2-1 subunit into the φPA3 map while retaining the light green subunit to preserve the N-terminal tail interaction. The N-terminal helix is positioned in the same pocket as that of the blue φPA3 subunit. **c** p2 φPA3 model with the corresponding φPA3 density (blue) and compared to the 201φ2-1 tomography density (gray, EMD-25221[17]). **d** Rigid body fitting of the published p4 201φ2-1 tetrameric model (PDB 7SQR[17]) into the φPA3 density (blue) and 201φ2-1 tomography volume (gray, EMD-25221[17]). The model protrudes outside of the tomography volume. **e** 201φ2-1 model (PDB 7SQR[17]) after independent rigid body fitting of each monomer into the φPA3 density alongside the improved fit of the resulting model into the 201φ2-1 tomography density (EMD-25221[17]).

flexible domains[26]. Long tails are often used around symmetry points to stabilize the assemblies[26]. The observations reported here suggest that quasi-equivalence may also be mediated directly by flexible tails and loops, not only forming quasi-symmetric tetramers but also flexibly linking adjacent tetramers together.

Given the extensive interactions between tetramers mediated by these long tails and loops, adding new subunits during growth while maintaining a permeability barrier may require the transient formation of dislocations or perhaps the aid of helper host proteins, such as chaperones, as occurs during the disassembly of clathrin lattices[27]. The observation of small assemblies, similar to those found in vitro, attached to an isolated shell (Supplementary Fig. 1a) also raises the possibility of a phage nuclear shell growth mechanism driven by the incorporation of small, pre-formed assemblies rather than individual monomers. The regular spacing and close proximity of these assemblies may also point to the existence of specialized regions along the phage nucleus where subunit addition occurs. This could allow for rapid and efficient growth while minimizing the number of disruptions made to the phage nuclear shell lattice. Alternatively, these small assemblies may correspond to distinct protein complexes with quite different functional roles.

The flexibility of the C-terminal tail, availability of some variation in its binding to neighboring subunits, and potential for toggling between symmetric states is perhaps what enables the astounding diversity in both shape and size of the bacteriophage nuclei[4], especially considering they appear to be largely constructed from a single ~70 kDa protein across various jumbo phages[5]. Despite the high degree of structural conservation, PhuN proteins from different species do not appear to interact during co-infections[28], suggesting that there may be localized differences altering their rates of lattice incorporation or prohibiting interactions altogether. As differing N- and C-terminal tail lengths account for the primary structural differences between AlphaFold predictions of known PhuN proteins

(Supplementary Fig. 10), a close analysis of PhuN evolution from diverse jumbo bacteriophages will be instrumental in delineating mechanistic similarities and differences that underlay the ability of PhuN proteins to assemble into the micron-scale phage nuclei observed in vivo. Investigating these questions will provide a more thorough understanding of where these structures came from, how they nucleate and grow, as well as how we can co-opt them for both experimental and practical applications going forward.

## Methods

### 6XHisMBPGp53 construct and expression

An N-terminal maltose-binding protein (MBP) tag and *E. coli* codon-optimized φPA3 *gp53* gene were inserted into a pET15 backbone to create pESN6. pESN6 was transformed into BL21 Star (DE3) pLysS cells for expression. About 20 mL of an overnight in LB was grown with carbenicillin and chloramphenicol. 1 L of Terrific Broth (TB) growth media was inoculated with around 2% overnight or until the starting $OD_{600}$ reached 0.001. Cells were grown with 250 rpm nutation at 37 °C until an $OD_{600}$ of 0.5. Cells and the incubator were then cooled to 16 °C. Expression was induced with 1 mM IPTG and proceeded overnight at 16 °C with 250 rpm nutation. Cells were spun down the following morning at $3,000 \times g$ for 8 min and pellets were transferred to 50 mL conical tubes. These were flash-frozen and stored at −80 °C or used immediately for purification.

### Phage shell isolation

*P. aeruginosa* PA01 (wild type or expressing 6XHisMBP-gp53) were lysed at 60 min post φPA3 infection in NP40 Lysis Buffer pH 6.5 (50 mM Bis-TrisPropane, 150 mM NaCl, 0.5% NP40, 5% glycerol, 5 mM DTT, 20 ng/μl Lysozyme, 1 mM EDTA, and 1 mM EGTA) by Dounce homogenizer. The lysate was clarified at $16,000 \times g$ for 5 min and the insoluble fraction resuspended in Wash/Shell buffer pH 6.5 (20 mM Bis-TrisPropane, 150 mM NaCl, 1 mM DTT, 1 mM EDTA, and 2 mM

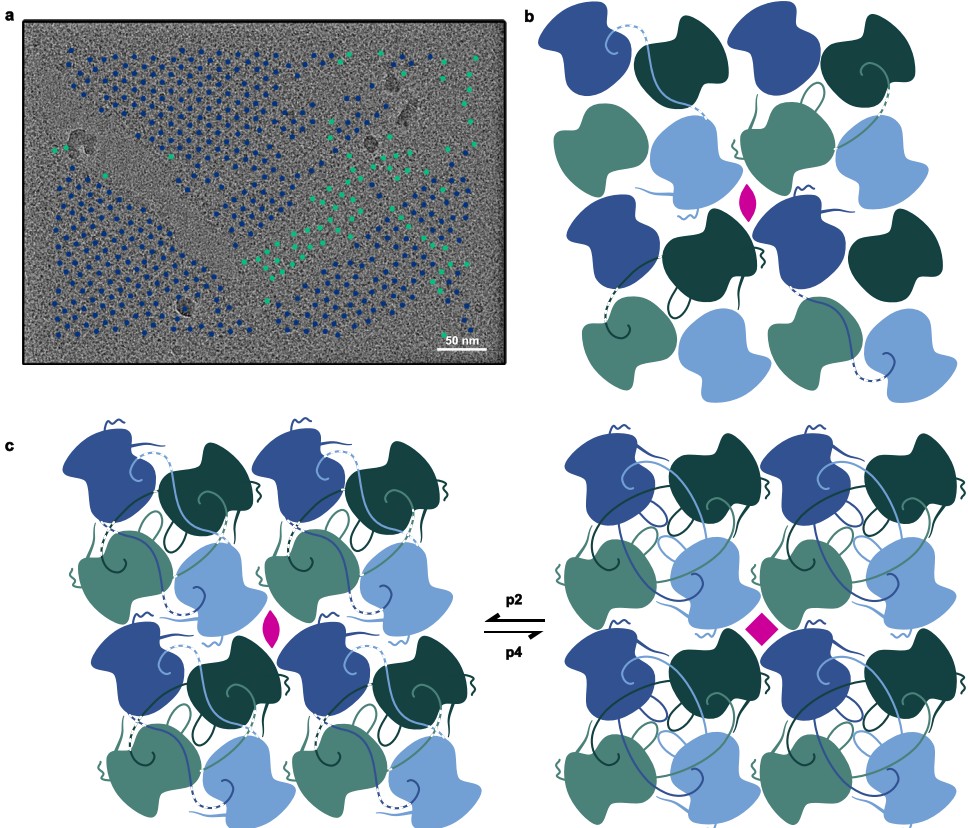

**Fig. 6 | PhuN assembles through a series of complex and likely adaptive C-terminal tail exchanges exhibiting both p2 and p4 symmetries. a** Micrograph of in vitro assembled PhuN lattices with the p2 (blue) and p4 (green) particles displayed. (*n* = 8). **b** Cartoon showing core tetramer with N- and C-terminal tail interactions as well as the loops visible clearly in 2D classes. **c** Cartoon model tracing the N- and C-terminal tail interactions at all four unique interfaces compared to the p4 equivalent.

MgCl$_2$). This was subject to further 500×*g* (5 min) and 15,000 × *g* (10 min) centrifugation with the insoluble fraction isolated and resuspended in Wash/Shell buffer each time. The final 15,000 × *g* pellet resuspension was used directly for downstream analysis by negative stain or cryo-EM.

## Immunofluorescence microscopy

Immunofluorescence microscopy samples were prepared as previously published and reiterated below[3]. About 5 mL overnight cultures of bESN30 (PA01 expressing 6XHisMBPGp53) were grown at 30 °C in LB media with gentamicin. A 1:30 back-dilution of the overnight culture:LB was grown at 30 °C for 1 h. Protein expression was induced with 0.1% arabinose. After 1 h of expression, an aliquot of uninfected cells was fixed while the remaining cultures were infected with ϕPA3 using MOI 1.5. Infected cell aliquots were collected and fixed at 60 mpi.

Samples were fixed with a 15 min incubation at room temperature in 5X Fix Solution (12.5% paraformaldehyde, 150 mM KPO4 pH 7.2) followed by an additional 20 min on ice. Samples were then washed in PBS three times and finally resuspended in GTE pH 7.65 (50 mM glucose, 10 mM EDTA, pH 8.0, and 20 mM Tris-HCl) with 10 μg/mL lysozyme. Resuspended cells were transferred to polylysinated coverslips and dried. The dry coverslips were incubated for 5 min each in cold methanol and then cold acetone. Cells were rehydrated by rinsing in PBS and incubating in PBS + 2% BSA blocking solution for 3 min. Cells were incubated with a 2 μg/mL dilution of primary antibody (Maltose Binding Protein Epitope Tag Antibody, Rockland Inc. Rabbit Polyclonal, Product # 200-401-385) in PBS + 2% BSA for 1 h and rinsed with 3, 7 min washes in fresh PBS + 2% BSA. Coverslips were then incubated in the dark for 1 hr with secondary antibody (Thermo Fisher Goat Anti-rabbit IgG (H + L) Highly Cross-Adsorbed Secondary Antibody, Alexa Fluor555, Catalog #A-21429) diluted to 4 μg/mL in PBS + 2% BSA. DAPI was added for the final 10 min of the incubation. Samples were washed in PBS three times for 7 min. Coverslips were then placed on slides using mounting media (v/v 90% glycerol, v/v 10%Tris pH 8.0, and w/v 0.5% propyl-gallate), sealed with clear nail polish, and imaged using a Zeiss Axiovert 200 M microscope.

## Fluorescence microscopy

Bacteria were prepared for imaging as previously described and reiterated below[4]. 1.2% agar pads prepared on concavity slides were supplemented with 0.05% arabinose to induce protein expression and 1 μg/mL DAPI for phage nucleus staining. About 5 μL of resuspended *P. aeruginosa* strain PAO1 colonies expressing PhuN-mNeonGreen deletions were grown on the agar pad at 30 °C for 3 h in a humid chamber. About 3 μL of ΦPA3 phage lysate (10⁹ PFU/mL) was spread on the agar pad and allowed to infect for 30 min at 30 °C in the humid chamber before imaging.

Microscopy was performed on an inverted epifluorescence (Ti2-E, Nikon, Tokyo, Japan) equipped with a Photometrics Prime 95B 25-mm camera and the Perfect Focus System (PFS). Images were acquired using Nikon Elements AR software (version 5.02.00). Cells were imaged through channels of phase contrast (200 ms exposure, for cell recognition), blue (DAPI, 50 ms exposure, for phage DNA), and green (GFP, 200 ms exposure, for PhuN-mNeonGreen constructs) at 100x objective magnification. For each condition, cells were imaged in 17 stacks in the Z-axis centered on the middle focal plane with a step size of 0.26 μm. Images were deconvolved with Huygens Essential

20.10 (Scientific Volume Imaging, the Netherlands; http://svi.nl) using the CMLE algorithm in the Deconvolution Wizard, and final figure images were prepared in Fiji[29].

### Purification of 6XHisMBPGp53 and all deletions

A cell pellet from 1 L of culture was resuspended in 12 mL of Lysis Buffer pH 7.42 (20 mM HEPES, 500 mM NaCl, 0.2% TritonX100, 1 mM DTT, 5% Glycerol, and 5 mM $MgCl_2$) per each gram of pellet with 1 cOmplete Protease Inhibitor Tablet and benzonase. This mixture was homogenized using a manual glass homogenizer and lysed using an Avestin EmulsiflexC3. The sample was clarified with a 20 min spin at $18,500 \times g$. The soluble fraction was nutated with equilibrated Ni·NTA resin for 1–2 h in the cold room. The sample was transferred to a gravity column and washed 2X with ATP Wash Buffer pH 7.42 (5 mM ATP, 5 mM $MgCl_2$, 20 mM HEPES, 500 mM NaCl, 20 mM Imidazole, 1 mM DTT, and 5% Glycerol) and 2X with Wash Buffer pH 7.42 (20 mM HEPES, 500 mM NaCl, 20 mM Imidazole, 1 mM DTT, and 5% Glycerol). Finally, the sample was eluted and collected in the Elution Buffer pH 7.42 (20 mM HEPES, 150 mM NaCl, 600 mM Imidazole, 1 mM DTT, and 5% Glycerol). The eluate was aliquoted and flash-frozen the same day and stored at −80 °C.

### Anion exchange

The sample (thawed or fresh) was buffer exchanged into MonoQ Binding Buffer (pH 7.65: 20 mM Tris, 50 mM NaCl, 1 mM DTT, 0.5 cOmplete Protease Inhibitor Tablet; pH 6.5: 20 mM Bis-TrisPropane, 50 mM NaCl, 1 mM DTT, 1 mM EDTA, and 0.5 cOmplete Protease Inhibitor Tablet) using 3.5 K MWCO Snakeskin dialysis tubing or a 7 K MWCO Zeba Spin Desalting Column. The sample was loaded on the 10/100 or 5/50 GL MonoQ Anion Exchange Column, washed, and eluted over a linear gradient from 0 to 65% over 13 mL (5/50 gL) using the Elution Buffer (pH 7.65: 20 mM Tris, 1 M NaCl, 1 mM DTT, 0.5 cOmplete Protease Inhibitor Tablet; pH 6.5: 20 mM Bis-TrisPropane, 1 M NaCl, 1 mM DTT, 1 mM EDTA, and 0.5 cOmplete Protease Inhibitor Tablet)

### Negative stain electron microscopy preparation and imaging

About 3 μL of the sample were applied to a glow-discharged carbon-coated grid (30 s, 20 mA). After 30 s, this was flushed away with deionized water followed by 30 μL uranyl formate (UF). Approximately 5 μL UF were left on the grid to stain for 30 s. This stain wash and incubation was repeated two more times. The grid was finally dried by filter paper side-blotting and gentle wafting through the air. The grids were imaged using a 120 kV FEI Tecnai T12 (LaB6 filament, Gatan UltraScan 895 4k CCD) or 200 kV FEI Tecnai T20 (FEG electron source, TVIPS 8 K × 8 K camera). Images were collected at 62 kx (defocus ~ −0.981 nm) or 145 kx (defocus ~ −0.295 nm).

### Cryo-EM imaging of isolated shell fragments

6XHisMBP-PhuN isolated shell fragments were imaged using the 200 kV Glacios microscope. About 34 micrographs were collected manually, six of which were used to pick and extract 9,568 particles. After three rounds of classification, 376 particles remained in the class shown in the Supplementary Fig. 1b inset. The WT PhuN isolated shell fragments were imaged using the 200 kV Talos Arctica (GATAN K3 camera). 10,864 particles were selected from 36 micrographs and processed to a final p2 class with 3,758 particles and p4 class with 1,538 particles (Fig. 1a).

### 2D crystal preparation and sample freezing

The most concentrated monomeric fraction was selected from the Anion Exchange purification. The concentration was used to calculate the volume of buffer and sample needed to reach a working concentration of 0.01 mg/mL 6XHisMBPgp53 in a 30 μL crystallization drop. A humidity chamber was made by wetting filter paper inside a petri dish. A Teflon block with approximately 5 mm wide and 13 mm

deep wells was placed in this chamber. The calculated volume of 2D Crystallization Buffer pH 6.5 (20 mM Bis-TrisPropane, 150 mM NaCl, 1 mM DTT, 1 mM EDTA, 2 mM $MgCl_2$, 1 mM ATP, and 0.25 cOmplete Protease Inhibitor Tablet) was pipetted into each well. 1 μL of Lipid Mix (0.73 mg/mL EGG PC and 0.27 mg/mL DGS-NTA(Ni)) was dispensed onto each buffer droplet. The volume of concentrated protein was then gently added to each droplet. After a 1 h incubation at room temperature, the lattices were adsorbed by directly touching the grid surface to the lipid monolayer using either amino-functionalized graphene oxide[20] or QUANTIFOIL R1.2/1.3 400-mesh copper holey carbon grids without glow-discharging. These were then plunge-frozen using a Vitrobot Mark IV (FEI) at 10 °C with 100% humidity using a 4 s blot time, 0 s wait time, and blot force of 0 for the functionalized grids and a 6 s blot time, 0 s wait time, and blot force of 3 for the QUANTIFOIL grids.

### Acquisition of cryo-electron microscopy data

As the data is composed of 15 collections with varying tilt, grid type, and frame rates, the following description and Supplementary Table 2 summarize the ranges of settings for all of the collections combined. The micrographs were collected using two 300 kV FEI Titan Krios electron microscopes equipped with GATAN K3 direct electron detectors and operated using SerialEM software. A nominal defocus range of 0.8–2.5 μm under focus was used for collection. A nominal magnification of 105 kx, corresponding to a physical pixel size of 0.835 or 0.834 Å, was used. A total dose of 66 or 67 e-/$Å^2$ was maintained for each collection. The samples were tilted from 0° to 60° for static dataset collection to obtain side-views of the 2D crystalline sample. To mitigate beam-induced motion, we utilized $GO/NH_2$ coated R1.2/1.3 300 mesh gold grids[20] and collected data using a higher frame rate to obtain a finer sampling of the beam-induced motion.

### Processing cryo-electron microscopy data

All micrographs were motion-corrected using UCSF MotionCor2[22] with a 9 by 7 patch. Each tilt, frame rate, and grid type was imported and processed separately through 2D classification.

The first datasets, collected on Quantifoil grids, were used to generate the first 3D model. After importing the motion-corrected micrographs into cryoSPARC[30], CTF estimation was done using cryoSPARC Patch CTF estimation (multi). cryoSPARC Blob Picker was then used to initially select particles. Two rounds of 2D classification were used to clean up picks. The selected particles from each tilt (0° 2,123 ptcls, 40° 308,819 ptcls, 50° 194,876 ptcls) were merged in a cryoSPARC Ab-Initio Reconstruction with C1 symmetry. A 176,652 particle class from the Ab-Initio Reconstruction gave a clear lattice volume with an apparent p2 symmetry. This volume was further used as a template for cryoSPARC Heterogenous Refinement.

Additional datasets underwent the same processing through 2D classification. Selected particles were added to the best particle subset for cryoSPARC Heterogenous Refinements with the best-resolved volume as a template. The template was ultimately recentered on the diamond-shaped channel at which point p2 symmetry was also imposed going forward. The centered template was also used to create a particle-picking template for each tilted dataset. The template was generated using the relion project feature in RELION v.3.0.8[31], saving a projection ±5° from the tilt at data collection every 3° and repeating this every 3° around the z-axis from 0–180°. This yielded 244 templates for each tilted dataset. These templates were imported into cryoSPARC and all datasets were repicked using their respective template set. The picks were cleaned up with two rounds of 2D Classification resulting in ~600,000 particles. These better-centered particles were classified using cryoSPARC Heterogenous Refinement and the resulting best class was processed using cryoSPARC Non-Uniform Refinement[32] to a reported 4.11 Å. These particles were exported for further processing in cisTEM[33,34]. In cisTEM, a mask was applied to the central tetramer for several rounds of Manual 3D Refinement[33]. The

reported resolution estimates did not always correspond to changes in map quality, thus each map was visually inspected for improvement. The particles were further classified in 3D using cisTEM, resulting in a final class containing 29,303 particles[33]. This final volume was deconvolved to mitigate stretching along the z-axis due to the missing cone of information resulting from tilted data collection.

### Deconvolution of a Map with Preferred Orientation
To help minimize the blurring due to limited tilt angles and subsequent missing cone of Fourier data, deconvolution was performed using ER-Decon II. This software was initially developed to deconvolve 4D wide-field light microscopy volumes with extremely low signal-to-noise ratio[35], but has subsequently been applied to STEM[36] and TEM[37] cryo-EM data. ER-Decon II is implemented in the program PRIISM[38]. To apply ER-Decon II on a cryo-EM density map, an optical transfer function (OTF) that characterizes the effects of preferred orientation and is equivalent to the point spread function (PSF) is generated from the directional FSC of the cryo-EM map which provides a representation of preferred orientation[39]. The smoothing and nonlinearity parameters for deconvolution were 0.5 and 10,000, respectively. The detailed procedure and full evaluation of cryo-EM volume deconvolution will be described in detail separately, but the general principles have been addressed previously by ref. [37].

### Model building
RoseTTAFold[15] and AlphaFold[16] were utilized to generate initial models. RoseTTAFold provided us with a first look at PhuN Gp53 while AlphaFold ultimately gave us a model which fit into the density slightly better, particularly after processing with RosettaRelax[40]. The N-terminus was positioned manually in ChimeraX Isolde[41] and further refined using RosettaRefine[40]. Residues 1–18, 276–287, and 556–602 did not have corresponding densities and were deleted for fitting. Invading C-terminal tails were similarly placed using Chimera[42], relaxed using ChimeraX Isolde[41,43], and refined using RosettaRefine[40]. The final model fitting was done using our deconvolved map (low pass filter 4.25 Å, B-factor 60 Å² applied in Relion), fitting each asymmetric monomer within the dimer density and doing further refinements using both RosettaRefine[40] and ChimeraX Isolde[41]. The dimer was used to create a p2 symmetric tetramer and the interface between the dimers was refined using ChimeraX Isolde[41]. The two subunits making this interface were duplicated and "fit in map" in ChimeraX to create the final p2 symmetric tetramer model[42,43]. Structure validation was done using Phenix MolProbity[44,45].

The larger, 16-mer assembly layer was constructed using a map processed with a larger mask (EMD-29451, Tracing p2 phiPA3 PhuN Tetramer Interfaces), resulting in a lower resolution, as a guide. This map included neighboring subunits around the core tetramer. Copies of the higher resolution PhuN tetramer map (EMD-29310, Deconvolved phiPA3 PhuN Tetramer, p2) were fit into the lower resolution map using the "Fit in Map" tool in ChimeraX. The corresponding 16-mer model was built by rigid body fitting copies of tetramer models into the newly positioned higher resolution tetramer maps with the ChimeraX "Fit in Map" tool.

### Data analysis and figure preparation
Figures were created using UCSF Chimera[42] and ChimeraX[43] using the deconvolved tetramer map (low pass filter 4 Å, B-factor 80 Å² applied in Relion)[43]. 2D class panels were prepared in 3dmod[46] Version 4.11.20 using the Low-frequency sigma (0.050) as well as the high-frequency cutoff (0.450) and falloff (0.027) parameters. EM micrograph panels were also prepared in 2dmod[46]. Anion exchange traces were plotted using MatLab (MATLAB. (2021,2022). *version 9.11.0 and 9.12.0 (R2021b and R2022a)*. Natick, Massachusetts: The MathWorks Inc. Figures were assembled in Affinity Designer (1.10.5). Electrostatic potential coloration was completed using the APBS and PDB2PQR Biomolecular Solvation Software web server at pH 6.5[47].

### Reporting summary
Further information on research design is available in the Nature Portfolio Reporting Summary linked to this article.

## Data availability
The cryo-EM maps generated in this study have been deposited to the Electron Microscopy Data Bank (EMDB) under accession codes EMD-29550 (phiPA3 PhuN Tetramer, p2), EMD-29310 (Deconvolved phiPA3 PhuN Tetramer, p2), and EMD-29451 (Tracing p2 phiPA3 PhuN Tetramer Interfaces). The atomic coordinates have been deposited to the PDB under accession codes 8FNE (phiPA3 PhuN Tetramer, p2) and 8FV5 (Representation of 16-mer phiPA3 PhuN Lattice, p2). Publicly available entries used in this study are PDB 7SQR and EMDB EMD-25221. All other data, bacterial strains, and plasmids are available from the corresponding author on a reasonable request.

## Code availability
PRIISM and ER-Decon II are both available on request from agard@msg.ucsf.edu as the terms of the license established prior to this study with the University of California and HHMI do not allow uploading the code in a public repository.

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

## Acknowledgements

We thank past and present members of the Agard laboratory for many helpful discussions and support throughout this project; particularly, Feng Wang for functionalized EM grids and Shawn Zheng for devel-oping *Motioncor2*; David Bulkley, Eric Tse, Glenn Gilbert (W.M. Keck Foundation Advanced Microscopy Laboratory at UCSF, Mission Bay), and Alexander Myasnikov (current: DCI-Lausanne) for maintaining the cryo-EM facility and assistance with data collection; Matt Harrington and Joshua Baker-LePain for computational support with the UCSF Wynton Cluster; Amanda Drennan (Rayment Laboratory, UW-Madison) for teaching ESN how to prepare 2D crystals. Molecular graphics and analyses were performed with UCSF Chimera (NIH P41-GM103311) and UCSF ChimeraX (NIH R01-GM129325); This research was supported by: NIH grant R35GM118099 (D.A.A.) and NIH facilities grants 1S10OD026881 (D.A.A.), 1S10OD020054 (D.A.A.), 1S10OD021741 (D.A.A.), Microsoft (M.B. and D.B.), Open Philanthropy and HHMI (D.B.), the Washington Research Foundation (M.B.), NIH grant R01GM127489 and R01AI171041 (J.B.-D.), UCSF Program for Breakthrough Biomedical Research funded in part by the Sandler Foundation (J.B.-D.), NIH R35GM140847 and HHMI (Y.C.).

## Author contributions

E.S.N. performed cloning, experimental design, in vitro purification, sample preparation, EM data collection and analysis, model fitting, coordinated experimental design for fragment isolations with M.M.-M., trained M.M.-M. in negative stain EM, and prepared this manuscript; all under the supervision of D.A.A. A.F.B. conducted motion correction of collected cryo-EM data and advised EM processing, experimental design, and interpretation. M.M.-M isolated shell fragments. C.K. carried out live fluorescence microscopy experiments from cloning through data analysis under the supervision of J.B.-D. M.B and D.B. provided early access to RoseTTAFold. J.L. and Y.C. conducted EM map decon-volution and resolution estimation.

## Competing interests

J.B.-D. is a scientific advisory board member of SNIPR Biome, Excision Biotherapeutics, and LeapFrog Bio, as well as a scientific advisory board

member and co-founder of Acrigen Biosciences. The Bondy-Denomy lab receives research support from Felix Biotechnology. The remaining authors declare no competing interests.
