## [Peer Review File · Nature Communications]

The Φ PA3 Phage Nucleus is Enclosed by a Self-Assembling 2D Crystalline LatticeREVIEWER COMMENTS

Reviewer #1 (Remarks to the Author):

This manuscript presents a preliminary structural analysis of a self-assembling protein layer that forms the shell of the presumptive phage nucleus in giant bacteriophage. This is a fascinating system just coming to light. The current work is an impressive example of hybrid structural analysis with AlphaFold structure prediction and cryo-EM reconstruction playing central roles. The structural studies are carefully analyzed and demonstrate high expertise as expected. The writing and illustrations are clear. This is important work worthy of publication in a competitive journal, once some concerns are addressed. Two main concerns are noted below. The first is technical regarding description of layer vs point symmetries. The second is a general need to be more cautious about what conclusions can be reached with certainty at this somewhat preliminary stage of analysis.

Main points:

1) The protein layer analyzed here appears to have a lattice that has nearly square dimensions but whose near-4-fold symmetry is broken down to 2-fold symmetry. The language used to describe the lattice geometry, symmetry and point symmetry needs to be cleared up. C2 is a point symmetry and should not be used to describe lattice symmetry. Everywhere lattice symmetry is being discussed, this should be described as p2 lattice symmetry. [A p2 lattice is one of several lattice types (including p4, p2121, p6, etc.) that supports 2-fold axes of symmetry.]

A subtler point arises concerning the lattice shape. While this is described as square, that distinction should be reserved for a lattice that obeys true p4 symmetry (or another lattice symmetry bearing 4-folds). Here, since the approximate p4 symmetry is broken, the lattice may be nearly-square or approximately square. Are the (oblique/parallelogram) unit cell lengths and angle given? I did not see the unit cell analyzed, unless I missed it. This is a fundamental feature of a repeating lattice and should be provided. Accurate measurement should be possible from direct imaging or from FFTs. How do the unit cell compare for the in vitro assemblies vs native layer fragments?

2) The authors' structural interpretations seem plausible and are probably correct at least at the level of detail possible at this stage. But my view is that the conclusions/assertions should be stated with more caution generally.

A] An example is the conclusion of which side of the layer faces inward towards the presumptive lumen of the phage nucleus. The authors reach their conclusion based on much smaller (non-layer) subunit assemblies that are definitely non-biological forms, also lacking the potentially driving effects of other cellular components, nucleic acids and the like. At this stage the sidedness is really a guess. This is fine, but a more circumspect phrasing is needed.

B] The use of nickelated lipid to drive the proteins to the air water interface is a nice trick for promoting a 2D protein layer/array (reminiscent of Dryden et al. Prot. Sci. 2009 in the related context of bacterial microcompartments). Can the investigators rule out that the position of the His tag on the protein does not have an important determining effect on how the subunits get arrayed in this non-biological setting? This should be brought up in the context of being cautious with interpretations. As noted above, a more explicit comparison of lattice dimensions between the in vitro and native cases would be helpful.

C] It seems the structure of the layer was created by manual placement of the predicted PhuN subunit structure? Given the investigator's exceptional level of computational expertise, this feels perhaps like a missed opportunity for a more objective/quantitative structural solution. Should it have been possible to solve the structure automatically by molecular replacement into the reconstructions? That would provide a measure of certainty in terms of the subunit arrangements defined. Are the top two solutions (under p2 symmetry with 2 molecules in the asu) much stronger than any subsequent solutions so that alternative interpretations can be eliminated? How many degrees are the two subunits in the asymmetric unit rotated relative to what would have been a 4-fold? What about the possibility of generating (minor) alternative models from AlphaFold? If that is possible, would the quantitative fit to EM maps allow the best predictive model to be identified experimentally? Etc.

Minor points:

- In describing the layer preparation and transfer process (line 393), a little more detail would be helpful. The layers were transferred from Teflon wells to grids by pipetting?

- Fig legend 1. "Borders" is misspelled.

Reviewer #2 (Remarks to the Author):

This work aims to corroborate the structural composition of the phage nucleus, a large structure induced by jumbo phage 201 that has originally been discovered by the same group, evidence of which they reported in the journals *Science*, *Cell* and several other publications. That structure was found by expression of gene products identified by mass spectrometry. GFP-fusion proteins of predicted structural proteins were expressed in *Pseudomonas*, characterized qualitatively in detail by using optical fluorescence light microscopy and visualized at low resolution by cryo electron tomography.

Here, they isolated phage nucleus structures based on homologs of the above structural protein family, which they define as phage nuclear enclosure (PhuN) proteins. They use a different jumbo phage, PA3, which infects *P.aeruginosa* and then determine structures from recombinantly expressed PhuN fusion proteins, showing that it forms 2D lattices in ice, by single particle cryo-EM averaging of lattice patches. They go on to describe these structures in detail and link pore-like features to their hypothesis of a growing phage assembly compartment.

The cryo-EM techniques include a number of innovative cryo-EM methods – in particular the deconvolution of the missing-wedge-induced stretching in the Z direction (which is unfortunately not described here).

The paper is well written and concise.

Major concerns

The main claim of the paper is that the reported structures in a 2D lattice represent the shell of the phage nucleus and that its porous structure explain properties such as diffusion of RNA and proteins.

However, evidence for this is mainly provided indirectly in the form of citations.

There is no in-situ structure.

There is no direct labeling, and densities for MBP are not evident in the map (ED Fig. 1d is entirely unclear). Why not label MBP with gold-conjugated antibodies?

No high resolution single particle structures from the particles in fractions I and II (Fig. 1b) have been attempted.

Short of collecting new in-situ CET data of PA3-gp53-MBP, why did the authors not simply add subtomogram averaging of existing in-situ CET data of 201-PhuN-GFP from their Science paper? This would allow validation of the general lattice parameters in-vivo, even with somewhat different unit cells. If GFP was too bulky and of concern for causing possible assembly artifacts, then so is the hexahistidine-MBP tag. Such charged or bulky affinity tags are indeed known to influence assembly patterns and they should therefore be removed before structural analysis, e.g. by engineering a cleavage site. Indeed, the authors state in the methods that MBP did not help affinity purification.

On line 65, the authors state "In this work, we demonstrate that Phage Nucleus fragments isolated from ϕ PA3 infected *P. aeruginosa* cells form a square lattice". Fig. 1a indeed shows square lattices. However, it is not clear that this lattice is a product of phage infection – as stated in the methods, a *P. aeruginosa* strain expressing MBP-gp53 was used and its purified products self-assembled into a 2D lattice on a functionalized lipid monolayer. Although phage was present, this expression of gp53 did not seem to depend on PA3. This statement is therefore somewhat misleading.

All molecular models were predicted by alphafold and fitted into the medium resolution deconvoluted cryo-EM maps – while this procedure is novel and represents an achievement on its own, the lattice interactions may depend on the used affinity tags and are ambiguous, as the authors point out themselves in the various possible patterns presented in Fig.4. The lattice is planar and no other assembly patterns have been considered. Overall, the conclusions about the roles of the putative pores and their dimensions are vague at best.

Minor concerns

Please add local map resolution figure.

No detailed description of the deconvolution method (to be published).

Please add dFSC after deconvolution.

Reviewer #1 (Remarks to the Author):

This manuscript presents a preliminary structural analysis of a self-assembling protein layer that forms the shell of the presumptive phage nucleus in giant bacteriophage. This is a fascinating system just coming to light. The current work is an impressive example of hybrid structural analysis with AlphaFold structure prediction and cryo-EM reconstruction playing central roles. The structural studies are carefully analyzed and demonstrate high expertise as expected. The writing and illustrations are clear. This is important work worthy of publication in a competitive journal, once some concerns are addressed. Two main concerns are noted below. The first is technical regarding description of layer vs point symmetries. The second is a general need to be more cautious about what conclusions can be reached with certainty at this somewhat preliminary stage of analysis.

Main points:

1) *The protein layer analyzed here appears to have a lattice that has nearly square dimensions but whose near-4-fold symmetry is broken down to 2-fold symmetry. The language used to describe the lattice geometry, symmetry and point symmetry needs to be cleared up. C2 is a point symmetry and should not be used to describe lattice symmetry. Everywhere lattice symmetry is being discussed, this should be described as p2 lattice symmetry. [A p2 lattice is one of several lattice types (including p4, p2121, p6, etc.) that supports 2-fold axes of symmetry.*

We thank the reviewer for spotting this inaccuracy. It has been corrected in the manuscript and all figures.

A subtler point arises concerning the lattice shape. While this is described as square, that distinction should be reserved for a lattice that obeys true p4 symmetry (or another lattice symmetry bearing 4-folds). Here, since the approximate p4 symmetry is broken, the lattice may be nearly-square or approximately square. Are the (oblique/parallelogram) unit cell lengths and angle given? I did not see the unit cell analyzed, unless I missed it. This is a fundamental feature of a repeating lattice and should be provided. Accurate measurement should be possible from direct imaging or from FFTs. How do the unit cell compare for the in vitro assemblies vs native layer fragments?

We have included an analysis of the unit cell lengths and angles in the text for both our in vitro assemblies and isolated fragments and have corrected our language where appropriate.

2) *The authors' structural interpretations seem plausible and are probably correct at least at the level of detail possible at this stage. But my view is that the conclusions/assertions should be stated with more caution generally.*

Thank you for highlighting this, we added additional data to support some of our interpretations and have modified our language to be more cautious with speculative statements in the text.

A] *An example is the conclusion of which side of the layer faces inward towards the presumptive lumen of the phage nucleus. The authors reach their conclusion based on much smaller (non-layer) subunit assemblies that are definitely non-biological forms, also lacking the potentially driving effects of other cellular components, nucleic acids and the like. At this stage the sidedness is really a guess. This is fine, but a more circumspect phrasing is needed.*

We have addressed this by using more cautious language and additionally citing a recently published study of a related system that is completely consistent with our proposed orientation. We included a direct comparison to the published in situ map (Fig. 5) and confirmed that our maps and models best fit into the now published cryoET data in the same orientation we initially suspected.

B] *The use of nickelated lipid to drive the proteins to the air water interface is a nice trick for promoting a 2D protein layer/array (reminiscent of Dryden et al. Prot. Sci. 2009 in the related context of bacterial microcompartments). Can the investigators rule out that the position of the His tag on the protein does not have an important determining effect on how the subunits get arrayed in this non-biological setting? This should be brought up in the context of being cautious with interpretations. As noted above, a more explicit comparison of lattice dimensions between the *in vitro* and native cases would be helpful.*

This is an understandable concern which we, too, shared initially. As no 6XHisMBP emerged in the reconstructions, we concluded that the 6XHisMBP was not ordered in our *in vitro* assembled arrays and thus unlikely to be influencing the assembly. To more directly address this point, in the revision we show that shell fragments isolated from an infection containing only WT, untagged PhuN (WT PA01 infected with WT ϕ PA3) produce lattices comparable to those assembled *in vitro*. This indicates that our His-MBP tag does not meaningfully alter lattice formation and that the *in vitro* assembled arrays are closely analogous to the ones assembled *in vivo*.

C] *It seems the structure of the layer was created by manual placement of the predicted PhuN subunit structure? Given the investigator's exceptional level of computational expertise, this feels perhaps like a missed opportunity for a more objective/quantitative structural solution. Should it have been possible to solve the structure automatically by molecular replacement into the reconstructions? That would provide a measure of certainty in terms of the subunit arrangements defined.*

We have expanded the methods to more accurately explain the approach taken to fitting both the tetrameric model and 16-mer layer. The key challenge with our sample was that the lattices had frequent imperfections—containing small patches of quasi-ordered tetramers with quite variable geometries and both p2 and p4 symmetries—limiting a more straightforward 2D crystallography approach or the meaningful collection of electron diffraction data which may have been passed by MR. Moreover, the project was well along with quite clear medium resolution 3D maps when AlphaFold or RosettaFold technologies became available. There were no other known homologous structures; hence, we used single particle approaches on tilted data to reconstruct 3D density maps which is substantially harder than conventional single particle analysis due to the limited views and tilt data quality. The end result was a map for the two molecules in the asu.

To summarize here, the predicted monomeric AlphaFold structure was used as a starting model to help interpret our density maps. Placement of the starting model into the density for both asymmetric monomers was unambiguous using the “Fit in Map” tool in Chimera. This tool computationally optimizes the local fit of atomic coordinates into the density map provided and is common practice with cryoEM maps and models. We then used RosettaRelax to adjust the two structures to better fit our higher resolution cryoEM density. Various loops and residues were further refined to fit and, in places, manually rebuilt into the higher resolution maps using Chimera, RosettaRefine, and ChimeraX ISOLDE. The resulting dimer was duplicated and fitted to build a full tetramer into the tetramer map. The interface between the dimers was refined. Finally, the two subunits forming the newly refined interface were symmetrized to create the final p2 symmetric tetramer.

The larger, 16-mer assembly layer has been updated using a lower resolution map (not shown, but will be deposited to the EMDb) that included neighboring subunits around the core tetramer. In short, copies of the higher resolution tetramer map were fit into the lower resolution map using the “Fit in Map” tool in ChimeraX. The 16-mer model was built by fitting copies of the tetramer models into the newly positioned higher resolution tetramer maps with the ChimeraX “Fit in Map” tool.

Are the top two solutions (under p2 symmetry with 2 molecules in the asu) much stronger than any subsequent solutions so that alternative interpretations can be eliminated? How many degrees are the two subunits in the asymmetric unit rotated relative to what would have been a 4-fold? What about the possibility of generating (minor) alternative models from AlphaFold? If that is possible, would the quantitative fit to EM maps allow the best predictive model to be identified experimentally? Etc.

The top five AlphaFold models converge on the same two domain structures. Aside from minor shifts, which we cannot meaningfully discern given our map resolution, the primary differences reside in the flexible N- and C-terminal tails whose conformations adapt to both build the tetramers and to link the tetramers together into the sheets. None of the AlphaFold predictions fit perfectly into the map so the highest scored model was built into the map with some guidance from the predictions on where to place the N-terminal helix. Building models into cryoEM maps is an iterative process, as described above, thus there aren't two clear solutions, but rather small improvements at every stage of the model building process.

Interestingly, the core domains of the two PhuNs within the asymmetric unit are nearly identical and, in fact, are rotated $\sim 90.14^\circ$ relative to each other (measured in Chimera). Our interpretation of the data is that the 2-fold arises largely from rotations of the entire core tetramer, perhaps mediated by the C-terminal tail and hairpin since they reside at these interfaces. There are likely minor shifts which we cannot reliably measure at our resolution.

Minor points:

- In describing the layer preparation and transfer process (line 393), a little more detail would be helpful. The layers were transferred from Teflon wells to grids by pipetting?

The layers were transferred by directly touching the carbon (regular Quantifoil) or GO (GO coated quantifoil) side of the grids to the lipid monolayer surface from the top. We have now added this information to the methods.

- Fig legend 1. "Borders" is misspelled.

Corrected!

Reviewer #2 (Remarks to the Author):

This work aims to corroborate the structural composition of the phage nucleus, a large structure induced by jumbo phage 201 that has originally been discovered by the same group, evidence of which they reported in the journals Science, Cell and several other publications. That structure was found by expression of gene products identified by mass spectrometry. GFP-fusion proteins of predicted structural proteins were expressed in Pseudomonas, characterized qualitatively in detail by using optical fluorescence light microscopy and visualized at low resolution by cryo electron tomography.

Here, they isolated phage nucleus structures based on homologs of the above structural protein family, which they define as phage nuclear enclosure (PhuN) proteins. They use a different jumbo phage, PA3, which infects P.aeruginosa and then determine structures from recombinantly expressed PhuN fusion proteins, showing that it forms 2D lattices in ice, by single particle cryo-EM averaging of lattice patches. They go on to describe these structures in detail and link pore-like features to their hypothesis of a growing phage assembly compartment.

The cryo-EM techniques include a number of innovative cryo-EM methods – in particular the deconvolution of the missing-wedge-induced stretching in the Z direction (which is unfortunately not described here).

The paper is well written and concise.

Major concerns

The main claim of the paper is that the reported structures in a 2D lattice represent the shell of the phage nucleus and that its porous structure explains properties such as diffusion of RNA and proteins.

However, evidence for this is mainly provided indirectly in the form of citations. There is no in-situ structure. There is no direct labeling, and densities for MBP are not evident in the map (ED Fig. 1d is entirely unclear). Why not label MBP with gold-conjugated antibodies? No high resolution single particle structures from the particles in fractions I and II (Fig. 1b) have been attempted.

Short of collecting new in-situ CET data of PA3-gp53-MBP, why did the authors not simply add subtomogram averaging of existing in-situ CET data of 201-PhuN-GFP from their Science paper? This would allow validation of the general lattice parameters in-vivo, even with somewhat different unit cells. If GFP was too bulky and of concern for causing possible assembly artifacts, then so is the hexa-histidine-MBP tag. Such charged or bulky affinity tags are indeed known to influence assembly patterns and they should therefore be removed before structural analysis, e.g. by engineering a cleavage site. Indeed, the authors state in the methods that MBP did not help affinity purification.

On line 65, the authors state "In this work, we demonstrate that Phage Nucleus fragments isolated from ϕ PA3 infected P. aeruginosa cells form a square lattice". Fig. 1a indeed shows square lattices. However, it is not clear that this lattice is a product of phage infection – as stated in the methods, a P. aeruginosa strain expressing MBP-gp53 was used and its purified products self-assembled into a 2D lattice on a functionalized lipid monolayer. Although phage was present, this expression of gp53 did not seem to depend on PA3. This statement is therefore somewhat misleading.

*We thank the reviewer for raising these concerns. In regards to the principal concern about the HisMBP solubility tag, we retained the MBP in our experiments as it reduces the protein's propensity for crashing out of solution, allowing for controlled assembly *in vitro*. We did not track the MBP as we could see our target protein clearly in our sample as a distinct structure from that of MBP. Since MBP was not visible in either the 2D averages or our 3D reconstruction and we could not find any literature indicating a propensity of MBP to form lattices or higher-order assemblies without precipitants, we*

conclude that the MBP, which is on a rather long 21 residue linker, is disordered in our lattices and, thus, unlikely to be driving the assembly of PhuN. To experimentally address these concerns, we have repeated the shell isolations using PA01 infected with WT ϕ PA3. The resulting WT PhuN isolates are essentially identical to the shell fragments isolated from infected *P. aeruginosa* expressing MBP-PhuN and to our *in vitro* MBP-PhuN lattices assembled using *E. coli* purified PhuN with lipid supports. This new data is now included in Figure 1 and fully supports all of our initial results and interpretations.

The previously published CET data were collected in collaboration with other labs which have since independently pursued, now published, similar questions in the 201 ϕ 2-1 system. While we do not determine an *in situ* structure, our isolated shell fragments (\pm MBP) corroborate our *in vitro* data, demonstrating with 2D class averages that the spacing, unit cells, and presence of both p2 and p4 symmetries are analogous to lattices grown *in vitro* with a 6XHisMBP tagged PhuN purified from *E. coli*. Moreover, the model determined from our *in vitro* assembled lattices fits exceedingly well into the now published (Laughlin, T.G., *et. al.* (2022)) *in situ* tomograms from the 201 ϕ 2-1 phage nuclear shell (Fig. 5). This both fully justifies our procedures and interpretations as well as demonstrates the high degree of structural conservation across jumbo phages.

All molecular models were predicted by alphafold and fitted into the medium resolution deconvoluted cryo-EM maps – while this procedure is novel and represents an achievement on its own, the lattice interactions may depend on the used affinity tags and are ambiguous, as the authors point out themselves in the various possible patterns presented in Fig.4. The lattice is planar and no other assembly patterns have been considered.

As no structure to date resolves the path of the C-terminal tail, we use Fig. 6 (previously Fig. 4) to highlight different pathways that the C-terminal tail may use to bridge across subunits in the large assemblies to enable the p2 and p4 symmetries observed in isolated and *in vitro* assembled lattices. Since both of these symmetries are present in the sample, there must be some means of converting between the two or determining which is adopted, an open question for future research. Other assembly patterns are not addressed because we do not observe them in the isolated samples from infected cells. Our results focus on the assembly of individual PhuN subunits into lattice sheets which we show are the biologically relevant form of PhuN in mature compartments. Questions regarding other assembly patterns—including differing curvatures and the range of small assemblies we observe in negative stain—are outside the scope of this single-particle focused work and would ultimately be better addressed with other methods and sample preparations.

Overall, the conclusions about the roles of the putative pores and their dimensions are vague at best. The discussion points about the roles of putative pores are not intended as conclusions but rather thought provoking hypotheses based on our solid structural observations. We feel this is an important point to address as it will be a large focus of further research regarding phage nuclear shell permeability. We use careful language to highlight that these are possible interpretations, not final conclusions.

Minor concerns

Please add local map resolution figure.

This has been added as Supplementary Figure 6.

No detailed description of the deconvolution method (to be published).

The principles behind the approach originally developed for optical fluorescence microscopy data (which has a missing cone governed by the finite lens NA) have been published. Likewise, the application to STEM and TEM cryo tomography data has been thoroughly described in two added citations. The exact application to coarsely sampled tilt data or, more generally, the preferred orientation problem in single particle samples will be published separately.

Please add dFSC after deconvolution.

The map-to-model FSC has been added to Supplementary Figure 6.

REVIEWERS' COMMENTS

Reviewer #1 (Remarks to the Author):

The authors have done a good job of clarifying some of the analysis and technical descriptions, while also qualifying some of the claims about points that remain open to interpretation.

One detail about layer symmetries still requires correction. On line 90 and in the legend to Figure 1, the authors give a unit cell angle of 88.2 degrees for the layer form described as p4. The unit cell angle for p4 is 90 degrees exactly. It is not a variable open to measurement. It is either 90 degrees, or else things are back to p2 with two molecules in the asymmetric unit instead of one. In the present case it seems most sensible to stick to the description as p4 with 90 degrees, with the understanding that an attempt to measure that angle is naturally subject to experimental error.

Reviewer #2 (Remarks to the Author):

This referee thanks the authors for their comprehensive revision of the manuscript. I appreciate the detailed response, added citations, and subtle change of language. In my opinion, all major concerns have been addressed.

I have no further concerns.

REVIEWERS' COMMENTS

Reviewer #1 (Remarks to the Author):

The authors have done a good job of clarifying some of the analysis and technical descriptions, while also qualifying some of the claims about points that remain open to interpretation.

We thank the reviewer for their constructive commentary on our manuscript.

One detail about layer symmetries still requires correction. On line 90 and in the legend to Figure 1, the authors give a unit cell angle of 88.2 degrees for the layer form described as p4. The unit cell angle for p4 is 90 degrees exactly. It is not a variable open to measurement. It is either 90 degrees, or else things are back to p2 with two molecules in the asymmetric unit instead of one. In the present case it seems most sensible to stick to the description as p4 with 90 degrees, with the understanding that an attempt to measure that angle is naturally subject to experimental error.

We have rounded the measured angles to more accurately represent the observed symmetries, given experimental error, and updated the corresponding text and figures.

Reviewer #2 (Remarks to the Author):

This referee thanks the authors for their comprehensive revision of the manuscript. I appreciate the detailed response, added citations, and subtle change of language. In my opinion, all major concerns have been addressed.

I have no further concerns.

We are glad we were able to address this reviewer's concerns. We thank the reviewer for their time and critiques of our manuscript.